# Effect of the Particle Size of Al/Ni Multilayer Powder on the Exothermic Characterization

**DOI:** 10.3390/ma13194394

**Published:** 2020-10-01

**Authors:** Shugo Miyake, Taisei Izumi, Rino Yamamoto

**Affiliations:** Department of Mechanical Engineering, Kobe City College of Technology, Kobe 651-2194, Japan; r112104@g.kobe-kosen.ac.jp (T.I.); r113177@g.kobe-kosen.ac.jp (R.Y.)

**Keywords:** self-propagating exothermic reaction, exothermic characterization, Al/Ni multilayer powder, particle size

## Abstract

In this study, the exothermic temperature performance of various Al/Ni multilayer powders with particle sizes ranging from under 75 to over 850 µm, which generate enormous heat during self-propagating exothermic reactions, was determined using a high-speed sampling pyrometer. The Al/Ni multilayer powders were prepared by a cold-rolling and pulverizing method. The multilayer constitution of the Al/Ni multilayer powders was examined by observing the cross-section of the powders using scanning electron microscopy; the results indicate that the powders had similar lamellar structures regardless of the particle size. Exothermic reactions were carried out to measure the temperature changes during the experiment using a pyrometer. We found that the maximum temperature and the duration of the exothermic reaction increased with an increase in the particle size caused by the heat dissipation of the surface area of the Al/Ni multilayer powder. This indicates that the thermal characteristics of the exothermic reaction of the Al/Ni multilayer powder can be controlled by adjusting the particle size of the Al/Ni multilayer powder. Finally, we concluded that this controllability of the exothermic phenomenon can be applied as a local heating source in a wide range of fields.

## 1. Introduction

Thermal damage in the production process for electric components and welding electrode connections is becoming a serious problem. This is because it gives excess heat to the whole of these materials by an electric and infrared radiation furnace. Thus, locally heating technology is necessary for these fields. One notable method is self-propagating exothermic materials. The self-propagating exothermic reaction has received increased interest in several fields due to its outstanding characteristics such as easy ignition, non-oxidation reaction, zero-emission, instantaneous reaction, and enormous heat generation [1,2,3,4,5]. These outstanding exothermic characteristics are due to the enthalpy difference between the materials and the formed intermetallic compounds before and after the reaction. Particularly, Al and Ni atoms achieve a desirable exothermic reaction with appropriately −60–70 kJ/mol of heat energy, and a reaction speed of 10 m/s [6,7,8,9]. The extraordinary features of the exothermic reaction of Al and Ni are beneficial for a wide range of practical applications such as the solder joining of silicon devices [10,11,12,13,14,15,16] and the initiation of neighboring injections [17,18,19]. For a smooth exothermic reaction, a stacked nanoscale-thick bilayer structure fabricated by a thin-film technology such as sputtering or plating techniques is required [20,21,22,23]. The sputtering method is a conventional and widely used technique for fabricating a nanoscale-thick bilayer structure because the exothermic heat can be adjusted by controlling the atomic concentration of Al and Ni atoms. However, several factors limit the practical application of the sputtering technique such as the very clean and flat surface of the substrate, and the requirement of large sputtering equipment installed in a vacuum vessel in a cleanroom. Thus, materials that can overcome these limitations are expected.

In a previous study, we fabricated an Al/Ni multilayer powder using the cold-rolling and pulverizing method and discussed its application [24,25,26,27]. The fabrication of an Al/Ni multilayer powder using these methods overcomes the disadvantages presented by the thin-film technology for the fabrication of an Al/Ni bilayer structure. The cold-rolling procedure is an industrial and conventional technique used for the mass production of materials. Moreover, the heat needed to fabricate powdered materials using this method can be easily controlled. In addition, since the cold-rolling procedure is based on the accumulative roll bonding technique with high reduction, the constructed rolled Al/Ni strip consists of sub-micrometer thick Al/Ni multilayers and cannot achieve the nanoscale thick bilayer fabricated by the thin-film process. Furthermore, we revealed that the maximum temperature of the Al/Ni multilayer powder during the exothermic reaction can be adjusted in the temperature range of 1300–1800 °C by adjusting the conditions of the cold-rolling technique [24,27]. In addition, we also found that the heat energy generated using this method can melt and join aluminum strips [25] and copper wires [26]. This indicates that the specific heating ability of the Al/Ni multilayer powder can be used as a heat source for open-space surface heating.

For practical use of the Al/Ni multilayer powder as a local heat source, a stable supply of materials with stable quality and efficient production are required. Hence, it is very important to know the influence of particle size on their exothermic characteristics as fundamental knowledge. Although some studies in the past have reported the intermetallic compound formation process of exothermic powder materials [28,29], there have been no reports on the kinetic relationship between exothermic characteristics and the range of the particle size of the Al/Ni multilayer powder during the self-propagation reaction. Moreover, using a static-thermal analysis method such as differential thermal analysis (DTA) and differential scanning calorimeter (DSC) to characterize the exothermic reaction, it is hard to understand whether the instantaneous exothermic characteristics are correctly revealed or not. Therefore, in this study, we attempted to directly estimate the relationship during the exothermic reaction. Generally, it is well known that the surface area per mass of the Al/Ni multilayer powder increased with a decrease in the particle size. Hence, the particle size may affect the exothermic temperature due to the heat transmission occurring above the powder surface.

In this paper, we report on the exothermic temperature performance of Al/Ni multilayer powders with different particle sizes ranging from under 75 to over 850 μm, measured using a high-speed sampling pyrometer to experimentally determine the instantaneous temperature change in the powder’s surface area.

## 2. Materials and Methods

Figure 1 shows the schematic of the fabrication procedure of the Al/Ni multilayer powder by the cold-rolling and pulverizing methods. Commercially available pure Al (>99.5%) and pure Ni (>99.9%) foils with sizes of approximately 50 × 50 mm^2^ were stacked alternately together to obtain a 1:1 atomic ratio of Al/Ni. The thicknesses of the pure Al and Ni foils were 0.030 and 0.015 mm, respectively. The total mass of the Al and Ni foils was almost 1 g. The stacked sheets were cold-rolled using a cold-rolling mill (Daito Seisakusho, DBR-70, Tokyo, Japan) consisting of two rolls with a diameter of 70 mm and a width of 115 mm. The meaningful conditions of the cold-rolling mill were a revolution speed of 45 rpm, rolling ratio of 50% (actual thickness ratio of each rolling), and the number of rolls was 40 passes. The rolling was repeated with high load and accumulative roll bonding until the stacked foils were thin multilayer sheets. The multilayer sheet was cut into appropriate sizes to insert a piece of the as-cold-rolled Al/Ni sheet in a pulverizer made of fine ceramics. The powdered Al/Ni materials were then classified using Japanese Industrial Standard sieves (Tokyo Screen CO. Ltd., Tokyo, Japan) with mesh sizes of 75, 250, 500, and 850 μm.

Exothermic temperature performance was characterized by radiation temperature measurements through exothermic experiments. Figure 2 illustrates a schematic of the experimental setup. A sample weighing 0.20 g of the Al/Ni powder was set into a depression (ø10 mm × 1 mm) on a heat-insulating brick, then the exothermic reaction was initiated using an electric spark. To generate electrical discharge in the Al/Ni powder, two electrodes with a potential difference were placed close to the Al/Ni powder, and a given electric energy was supplied to the Al/Ni powder. The electric power was supplied to the electrodes by a stabilized direct-current power supply 20 V-1 A. The exothermic temperature was measured using a pyrometer (Yamari Industries, IS310-MB23, Osaka, Japan) and a digital datalogger (Graphtec, GL-240, Kanagawa, Japan). The pyrometer consisted of a silicon photodiode with a temperature measurement range of 800–2300 °C and responsiveness of 10 ms. The sampling period of the datalogger was set to 100 Hz. The experimental atmosphere in the laboratory was kept at room temperature, controlled by an air-conditioner, and almost no temperature change around the experimental space was measured during the experiment. In addition, the states of the exothermic reactions were recorded using a digital camera every 0.1 s. These experiments were performed several times until the average tendency was confirmed, and variations in the experimental data are expressed in the sections below.

To investigate the microstructure and nanometer-scale elemental analysis of each fabrication stage, field emission scanning electron microscopy (FE-SEM: JEOL, JSM-7610F, Tokyo, Japan) and field emission electron probe microanalysis (FE-EPMA: JEOL, JXA-8530F Plus) were used. The proper and special specimens used for the cross-sectional observation were prepared as described below (but otherwise observed as prepared). First, the as-prepared powder sample was molded with epoxy resin and polished to be a mirror surface using several proper waterproof sandpapers. Then, cross-section polishing was performed using 5 kV of an argon ion beam (JEOL, IB-19520CCP) to achieve a more fine-smooth surface. Subsequently, the microstructure and elemental analysis were observed by FE-EPMA.

For the crystallographic analysis of the reacted Al/Ni powders, powder x-ray diffraction (XRD) and phase identification were conducted using x-ray diffraction equipment (RIGAKU, MiniFlex600, Tokyo, Japan) with Cu-k_α_ as the x-ray source and analysis software (RIGAKU, PDXL2), respectively.

## 3. Results

### 3.1. Microstructures of the Al/Ni Multilayer Materials

#### 3.1.1. Cold-rolled Al/Ni Multilayer Sheet

To investigate the constitution of the cold-rolled Al/Ni multilayer sheets after 40 passes, the cross-section of the sample was observed using FE-EPMA. Figure 3a shows the representative photo of the backscattered electron compositional image (COMPO image). A lamellar-like structure was confirmed in the samples in the axis parallel to the rolling direction. In addition, the average thickness of the individual layers was less than 1 µm. Figure 3b,c show the distributions of Al and Ni in the sample detected by a wavelength dispersion-type detector of the FE-EPMA. Al and Ni were clearly identified in the samples with the red-colored area, indicating the high-concentrated point of Al and/or Ni elements, and the blue-colored area, indicating the low-concentrated point. In addition, the Al and Ni were observed to exist alternately. However, the lamellar width of the Al/Ni layers observed by the dispersion-type detector of the FE-EPMA did not correspond to that observed by the COMPO image in Figure 3a. This was due to the difference in the spatial resolution of the detectors between the electron detection mode and the characteristic x-ray detection mode. As shown in Figure 3c, a highly concentrated domain of Ni was observed. Generally, the Ni phase is less deformed than the Al phase during the cold-rolling process. Thus, the domain of Ni that was not deformed remained in place.

Figure 4 shows the XRD pattern of the cold-rolled Al/Ni multilayer powders with mixed particle sizes from under 75 to over 850 µm (no classification). As shown in Figure 4, Al and Ni phases were identified. Here, if the NiAl intermetallic compounds are formed due to exothermic reaction during the cold-rolling process, the strong peak near 45° may include the peak of NiAl (110), in addition to Ni (111) and Al (200). Additionally, the peak near 65° may include the peak of NiAl (200) in addition to Al (220). In this case, strong peaks of NiAl (100) and (211) should be seen around 30° and 80°, respectively. However, these peaks could not be confirmed in the measurement. Furthermore, although the x-ray penetration depth of Cu-kα into the Al/Ni multilayer was assumed to be up to approximately 10 µm considering the absorption coefficients of Al and Ni, this result shows the non-reacted state due to the measurement of the randomized particles. In addition, the peak width of each diffraction pattern seemed to be wider, which was due to the effect of the cold-rolling procedure on the non-uniform strain of the lattice spacing.

#### 3.1.2. Pulverized Al/Ni Multilayer Powder

The classified powders with particle sizes under 75 and over 850 µm were observed by FE-SEM. Figure 5 shows the surface morphologies of the pulverized powders observed by a secondary electron (SE) detection mode of FE-SEM as representative images of the under 75 and over 850 µm powders. Similar stratum-like stacked layers were observed in both powders. This indicates that the pulverizing method had no effect on the thickness of the Al/Ni multilayers, but only made the particle size finer.

Figure 6 shows the cross-sectional COMPO images of the under 75 and over 850 µm classified powders. The lamellar structures of the two powders were similar and consisted of thin-multilayers, and the thickness of each layer was less than 0.1 µm. In addition, no voids or cracks were observed at the Al/Ni interfacial layer. This indicates that the pulverizing process had no effect on the Al/Ni multilayer interface.

### 3.2. Exothermic Characterization.

The representative exothermic reaction process captured by the digital camera extracted *N* = 2 or 3 experiments is shown in Figure 7 and Figure 8. Figure 7 shows the initiation stage up to 0.5 s every 0.1 s and Figure 8 shows the initiation stage up to 3.0 s every 0.5 s.

For the under 75 μm powder, the initiation flashed out immediately after the reaction and was over-exposed after 0.1 s, after which the reaction started to converge to finish after 0.3 s. In addition, the time at which the light emission progresses depends on the particle size. With an increase in the particle size, the elapsed time for the maximum emission increased, and the reaction did not converge even after 0.5 s. For the over 850 μm powder, a strong luminescence was observed over an elapsed time in comparison to the 250–500 and 500–850 µm powders, and the emission continued until 0.5 s, as shown in Figure 7. A comparison of the state of the powders several seconds after the reaction is shown in Figure 8. We observed that the emission peaks could be qualitatively recognized with an increase in time, and the intensity of the emission increased with an increase in the particle size. In addition, the electric spark in the moment of the reaction increased with an increase in the particle size.

Figure 9 shows the representative temperature curves measured by the pyrometer, which were extracted from *N* = 2 and more experiments, with a measurable range from 800 to 2300 °C due to the sensitivity of the silicon photodiode, as a function of the particle size. The temperature curve of the under 75 µm powders showed a single sharp peak of up to 1614 °C at 10 ms at the moment of the reaction. However, two or more additional peaks were observed, indicating that maximum temperatures were observed in the temperature curves of the other powders. Particularly, in the temperature curves of the powders between 75 and 250 µm and those over 850 µm, a notable sudden temperature drop and spike were observed. The maximum temperatures of the powders increased with increasing particle size. In addition, the duration time for the heat release increased with an increase in the particle size.

## 4. Discussion

First, we discuss the completeness of the exothermic reaction of the pulverized and classified the Al/Ni multilayer powders. This is because the effect of the particle size on the characteristics of the exothermic should be estimated without additional effects. Figure 10 shows the XRD analysis results of the under 75 µm (75 µm) and over 850 µm (850 µm) classified Al/Ni multilayer powders after the exothermic reaction. NiAl phase with a B2 crystallographic structure [21,30,31,32] was identified in both powders, confirming the formation of the NiAl intermetallic compound, and clear peaks of other intermetallic compounds have not been confirmed in the XRD analysis. As mentioned earlier, the enthalpy formation of the NiAl phase was larger than that of other intermetallic compounds of the Ni–Al system alloy [8,33,34]. Therefore, in this experiment, we confirmed that the conditions necessary to discuss the effect of the particle size on the exothermic characteristics were in place.

The exothermic reaction experiments confirmed the effect of the particle size on the maximum temperature and duration time of the exothermic reaction. Figure 9 shows the dependency of the maximum temperature on the particle size for the quantitative evaluation of the dependency of the duration time of the exothermic reaction on the particle size and the elapsed time from the time the reaction occurred to the time taken to reach a temperature point during heat release (1400, 1200, 1000, and 800 °C) as a function of the particle size was measured, as shown in Figure 11. Error bar shows 10% of deviation from the average value, which was estimated through *N* = 2 and more experiments and no lager error was recognized from our previous studies of 40 passes of cold-rolling [24].

The time duration of the under 75 µm powder was shorter than that of other particle sizes. In addition, the duration time of the exothermic reaction increased with an increase in the particle size. However, only a slight increase was observed among the 75 and 850 µm samples.

Herein, the dependency of the duration time of the exothermic reaction on the particle size was discussed based on heat transmission. Since the exothermic reaction was generated at room temperature and the heaped Al/Ni multilayer powder (a group of classified particles) was assumed to be a lump of powder with rough surface, the heat release from the heat source to the outside can be considered to be dominated by heat transmission. In this study, the effect of heat radiation was neglected as an exothermic reaction involves a phase transformation with a shrinkage of the volume of the materials and a change in the surface morphology at the moment of the reaction. Additionally, under an air atmosphere, the amount of energy by heat transfer is typically larger than that by heat radiation; therefore, we assumed that the effect of heat radiation is negligible. Moreover, the Biot number *Bi* = α·*L*/*λ* was not considered due to the small size and thickness of the powders.

Generally, heat transmission is expressed by Newton’s law using the following equation:*Q* = *A*·α·Δ*Τ*(1)
where *Q* is the amount of heat from the solid surface; *A* is the surface area; *α* is the local heat transfer coefficient; and Δ*T* is the temperature difference between the solid surface and atmosphere. Then, the value of *Q* is given by the enthalpy of formation and the mass of NiAl as follows:*Q* = Δ*H* = *H*_*NiAl*_ − (*H*_*Al*_, *H*_*Ni*_)(2)
where *H* is the enthalpy and Δ*H* is the enthalpy difference based on the binding energy of the atoms before and after the reaction. As above-mentioned, the enthalpy of formation of NiAl is the largest of the Al–Ni intermetallic compounds. As shown in Figure 10, all of the reacted Al/Ni multilayer powders become the NiAl phase, and thus the amount of heat *Q* is a constant value. In addition, the mass of NiAl for the experiment was equal. This indicates that the Δ*T* of Equation (1) is sensitive to the surface area. In addition, the area difference between the 75 and 850 µm powders when the shapes are assumed to be flake-like was estimated to be quite large and depended on the density. Furthermore, regardless of the accuracy of the measured surface area and density, the local heat transfer coefficient should be almost equal among these particles. Therefore, the maximum temperature is dependent on the particle size.

The dependency of the duration time on the particle size is interesting. Since the exothermic reaction is a self-propagating reaction, although the particles are small-sized, the reaction propagated from the reaction starting point into the particle. Moreover, as the larger particle should have a large heat capacity and propagation length inside the particle, inevitably, it takes a little time for the reaction and heat release process. To further understand the relationship between the particle size of the Al/Ni multilayer powder and its exothermic characterization, it is necessary to perform experiments on finely classified powders and utilize numerical simulation in future works.

## 5. Conclusions

In this paper, we report the exothermic temperature performance of Al/Ni multilayer powders with various particle sizes from under 75 to over 850 µm. FE-SEM and FE-EPMA analyses were carried out to observe the morphology and microstructures of the Al/Ni multilayer powder. The results revealed that the powders consisted of a fine bilayer structure of Al and Ni with no voids or cracks at the Al/Ni interface. The XRD results confirmed the formation of the NiAl intermetallic compound by the exothermic reaction. The results of the exothermic reaction experiments indicate that the maximum exothermic temperature and duration time of the exothermic reaction can be controlled by adjusting the particle size of the Al/Ni multilayer powder due to the heat dissipation of the surface area of the Al/Ni multilayer powder. These results will provide insights into improving future works on the practical use of exothermic materials as a novel heat source in practical industrial use.

## Figures and Tables

**Figure 1 materials-13-04394-f001:**
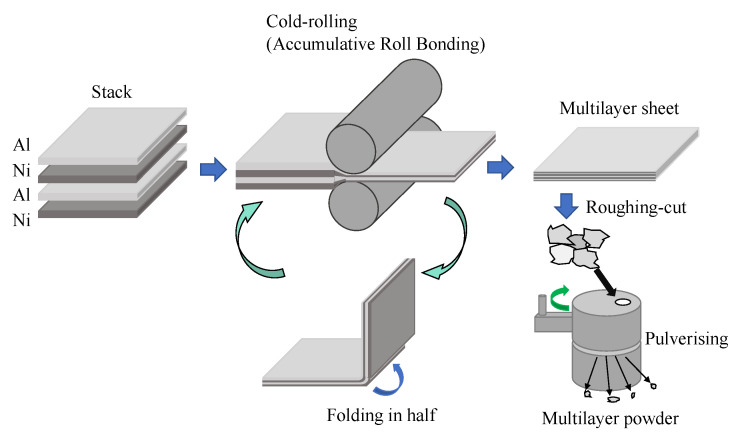
Schematic of the fabrication procedure of the Al/Ni multilayer powder by the cold-rolling and pulverizing methods.

**Figure 2 materials-13-04394-f002:**
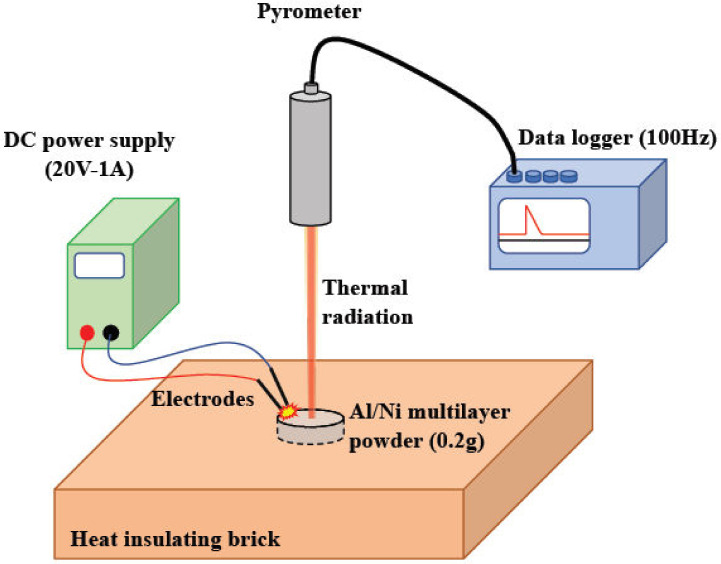
Experimental setup for the temperature measurement during the exothermic reaction by the high-speed pyrometer.

**Figure 3 materials-13-04394-f003:**
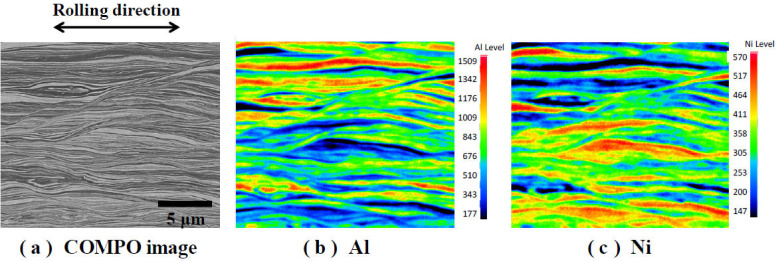
Cross-section observation of the structure of the Al/Ni multilayer sheet after 40 passes by the FE-EPMA. (**a**) photo of backscattered electron compositional image, (**b**) Al concentration, and (**c**) Ni concentration distributions analyzed by the wavelength dispersion type detector.

**Figure 4 materials-13-04394-f004:**
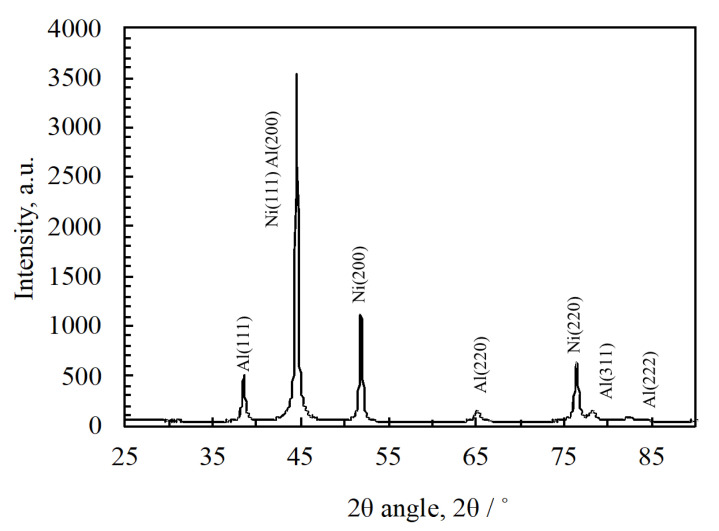
Result of the x-ray diffraction (XRD) measurement of the cold-rolled Al/Ni multilayer powder (before the reaction).

**Figure 5 materials-13-04394-f005:**
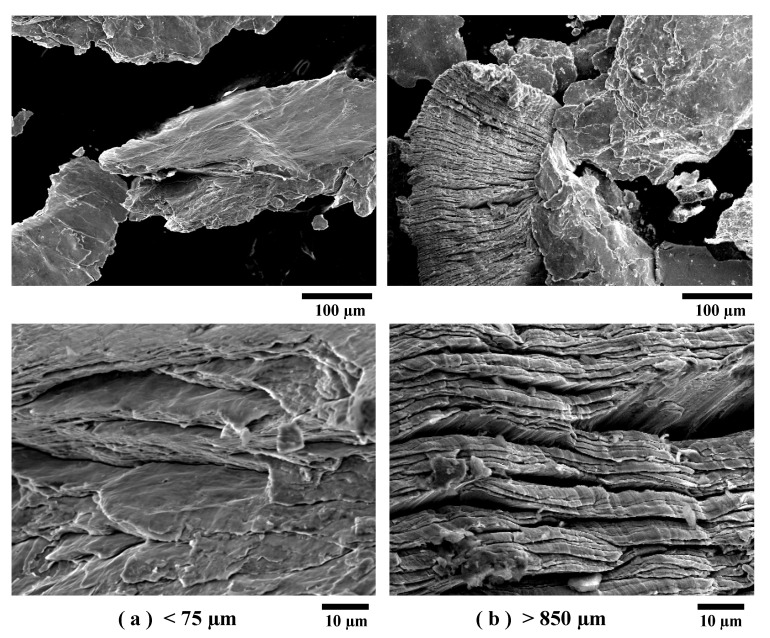
Morphologies of the (**a**) under 75 and (**b**) over 850 μm pulverized Al/Ni multilayer powders observed by a secondary electron (SE) detection mode of FE-SEM.

**Figure 6 materials-13-04394-f006:**
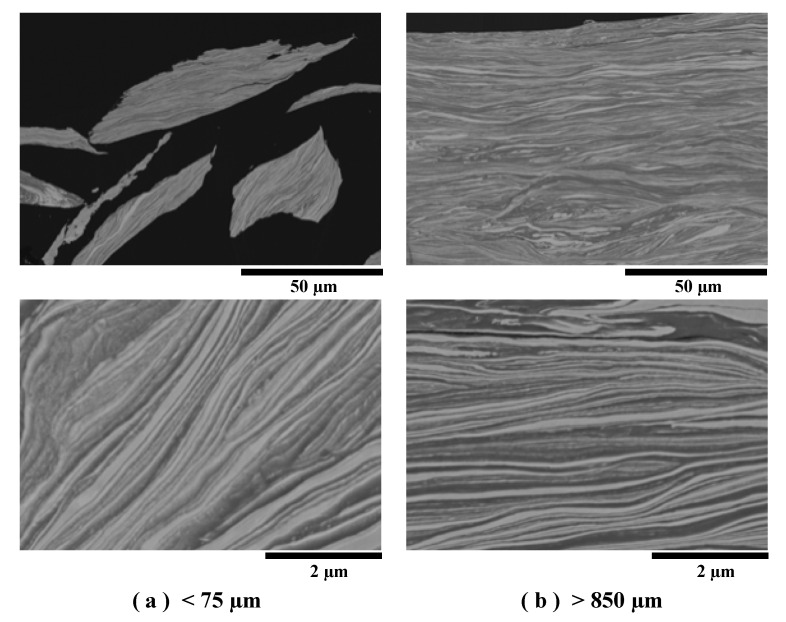
COMPO images of the (**a**) under 75 and (**b**) over 850 µm classified Al/Ni multilayer powders.

**Figure 7 materials-13-04394-f007:**
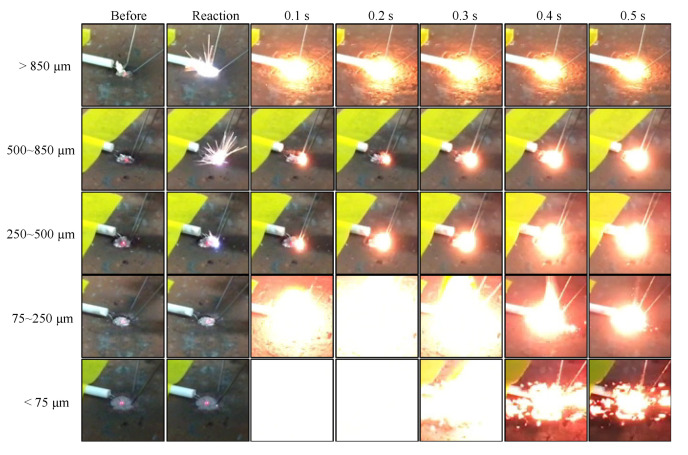
The exothermic reaction process recorded by the digital camera at the initiation stage up to 0.5 s at an interval of 0.1 s.

**Figure 8 materials-13-04394-f008:**
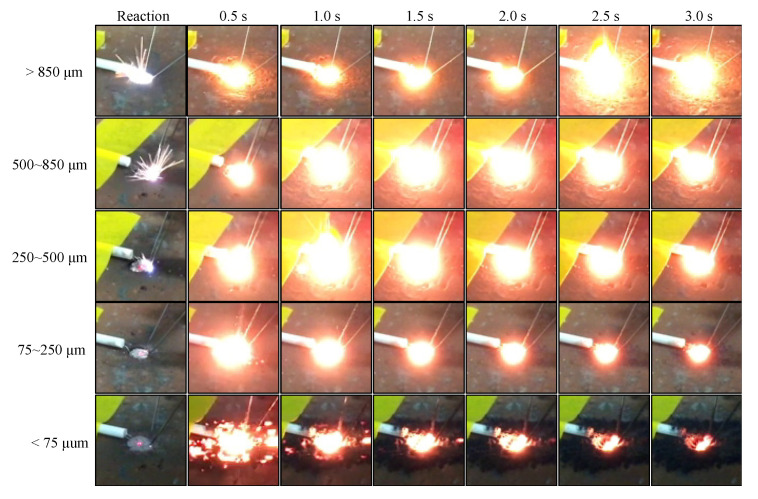
The exothermic reaction process recorded by the digital camera up to 3.0 s at an interval of 0.5 s.

**Figure 9 materials-13-04394-f009:**
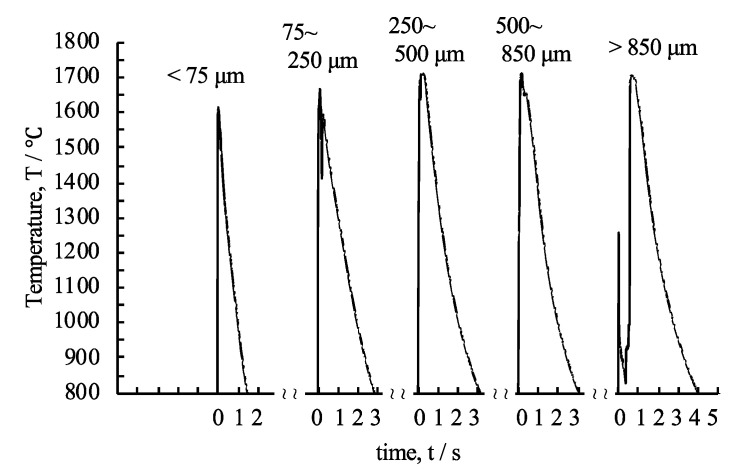
Temperature curves of the Al/Ni multilayer powders during the exothermic reaction measured by the pyrometer as a function of the classified particle sizes.

**Figure 10 materials-13-04394-f010:**
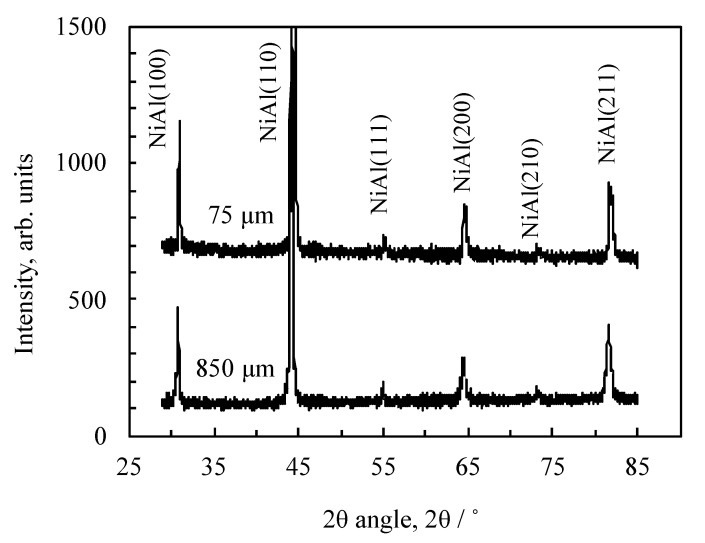
X-ray diffraction patterns under 75 and over 850 µm of the classified Al/Ni multilayer powder after the exothermic reaction.

**Figure 11 materials-13-04394-f011:**
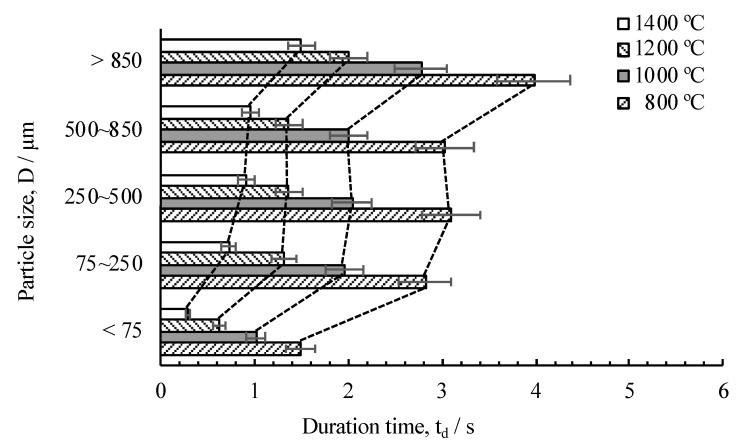
Evaluation of the dependency of the duration time of the exothermic reaction on the particle size as a function of increased temperature.

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
