# Peer review of "Effect of the Particle Size of Al/Ni Multilayer Powder on the Exothermic Characterization"

_materials, 2020, doi:10.3390/ma13194394_

Round 1

Reviewer 1 Report

In this manuscript, Miyake et al. reported an exothermic reaction results of Al/Ni multilayer powder with particle size ranging from under 75 um to over 850 um. The multilayer powders were then characterized by SEM and consistent lamellar structures independent of particle sizes were observed. The exothermic reaction was further characterized by digital camera, pyrometer, and XRD. The author also suggested that the material could be used for a wide range of industries with the controllable exothermic phenomenon. The manuscript needs extensive language revision. Also a few minor issues needs to be addressed before acceptance by a novel journal like Materials. Please see detailed comments below:

  1. Currently, the manuscript is hard to understand because of a great number of grammar errors. Can the author work on the manuscript to fix all the issues?
  2. How did the author tell the difference between XRD patterns of un-reacted Al /Ni and the Al/Ni multilayers? Specifically, (a) Ni (111) / Al (200) and NiAl (110) at ~45° (b) Al (220) and NiAl (200) at ~65°. Are there any other spectroscopic analysis results (eg. XPS or Auger electron spectroscopy) that support the formation of the Al / Ni multilayers?
  3. What is the n number for the results shown in Figure 11? Can the author add error bars to the plot?

Author Response

To Reviewer1
Thank you very much for your careful and fruitful reviewing. I deeply appreciate your suggestion and comments. We have revised the manuscript throughout. Please re-review the revised. Our replies to your pointed out are listed as follows:

Point 1: Currently, the manuscript is hard to understand because of a great number of grammar errors. Can the author work on the manuscript to fix all the issues?

Response 1:
I asked a native speaker to check the English of the manuscript. The revised manuscript is attached. Please review it again.

Point 2: How did the author tell the difference between XRD patterns of un-reacted Al /Ni and the Al/Ni multilayers? Specifically, (a) Ni (111) / Al (200) and NiAl (110) at ~45° (b) Al (220) and NiAl (200) at ~65°. Are there any other spectroscopic analysis results (eg. XPS or Auger electron spectroscopy) that support the formation of the Al / Ni multilayers?

Response 2:
The strong peak near 45 ° may include the peak of NiAl (110) in addition to Ni (111) and Al (200). Also the peak near 65 ° may include the peak of NiAl (200) in addition to Al (220). In this case, strong peaks of NiAl(100) and (211) should be seen around 30 ° and 80 °,respectively. However, these peaks could not be confirmed in the measurement. We added the explanations to the revised manuscript line 156.

Point 3: What is the n number for the results shown in Figure 11? Can the author add error bars to the plot?

Response 3:
I apologize to reviewer what it was a lack of explanation. I added error bars and comments in line239.

Reviewer 2 Report

The manuscript "Effect of particle size of Al/Ni multilayer powder on exothermic characterization" has been reviewed.

This is a technical paper describing the effect of particle size of AlNi multilayer powder on self-propagating exothermic reaction. 

The topic is not new, as evidenced also by many aged references (please add!).

English language must be deeply revised. Many sentences are convoluted and hard to understand.

The final goal of the experiments must be expanded and clarified, with particular attention to the novelty aspects in comparison to the state of the art.

After major revision can be reconsidered for publication.

Author Response

To Reviewer2
Thank you very much for your careful and fruitful reviewing. I deeply appreciate your suggestion and comments. We have revised the manuscript throughout. Please re-review the revised. Our replies to your pointed out are listed as follows:

Point 1:
The topic is not new, as evidenced also by many aged references (please add!).

Response 1:
Thank you for your pointed out what the topic of the manuscript is not new. However, we think it is new-knowledges due to following reasons.
Although some studies in the past have reported the intermetallic compound formation process of exothermic powder materials [28,29], there are no reports on the kinetic relationship between exothermic characteristics and the range of the particle size of the Al/Ni multilayer powder during the self-propagation reaction. Moreover, using static-thermal analysis method such as differential thermal analysis (DTA) and differential scanning calorimeter (DSC) to characterise the exothermic reaction, it is hard to understand whether the instantaneous exothermic characteristics is correctly revealed or not.
We added the above explanations to the revised manuscript in line 63 of introduction section.

Point 2:
English language must be deeply revised. Many sentences are convoluted and hard to understand.

Response 2:
I asked a native speaker to check the English of the manuscript. The revised manuscript is attached. Please review it again.

Point 3:
The final goal of the experiments must be expanded and clarified, with particular attention to the novelty aspects in comparison to the state of the art.

Response 3:
As you pointed out, the goal of the experiments is not clear. We added the purpose and goal of the study and experiments in introduction section of the revised manuscript. Please see the attached.

Reviewer 3 Report

The paper deals with well-known SHS reaction of Ni and Al particles, duration of SHS. Additional SEM observation of SHS product can be helpful and it can confirm the formation of the NiAl phase by EDS. Other intermetallics in small amount could be formed too.

Please, add spaces between number and unit. e.g.75 µm 

Line 131: Sentence: The X-ray source ...  It should be moved to part of Experiments.

Line 184 Figure 9: How do you explain the formation of two significant maximums for 75-250 and over 850 µm. My rocommendation is to perform DSC analysis if it is possible. It can confirm reaction duration and verify peak of maximum temperature despite the fact that another heating source is applied in DSC. The observation of reaction duration of Ni-Ti-Al powder mixture by DSC analysis and combination of heating in induction furnace and measuring of temperature by pyrometer. The results are stated in fpllowing paper: Salvetr, P.; Školáková, A.; Hudrisier, C.; Novák, P.; Vojtěch, D. Reactive Sintering Mechanism and Phase Formation in Ni-Ti-Al Powder Mixture During Heating. Materials 201811, 689.

Figure 10: SEM-BSE observation of reaction product can show other intermetallics in low amount.   

Author Response

To Reviewer3
Thank you very much for your careful and fruitful reviewing. I deeply appreciate your suggestion and comments. We have revised the manuscript throughout. Please re-review the revised. Our replies are listed as follows:

Point 1:
Please, add spaces between number and unit. e.g.75 µm

Response 1:
The point you pointed out has been revised in the attached manuscript.

Point 2:
Line 131: Sentence: The X-ray source ... It should be moved to part of Experiments.

Response 2:
The point you pointed out has been revised in the attached manuscript.

Point 3:
Line 184 Figure 9: How do you explain the formation of two significant maximums for 75-250 and over 850 µm. My rocommendation is to perform DSC analysis if it is possible. It can confirm reaction duration and verify peak of maximum temperature despite the fact that another heating source is applied in DSC. The observation of reaction duration of Ni-Ti-Al powder mixture by DSC analysis and combination of heating in induction furnace and measuring of temperature by pyrometer. The results are stated in fpllowing paper: Salvetr, P.; Školáková, A.; Hudrisier, C.; Novák, P.; Vojtěch, D. Reactive Sintering Mechanism and Phase Formation in Ni-Ti-Al Powder Mixture During Heating. Materials 2018, 11, 689.

Response 3:
Thank you for the pointed out what the topic of the manuscript is not new. However, we think it is new-knowledges due to the following reasons.
Although some studies in the past have reported the intermetallic compound formation process of exothermic powder materials as you pointed out [28,29], we think that, use the static-thermal analysis method such as DTA and DSC to characterise the exothermic reaction, it is hard to understand whether the instantaneous exothermic characteristics is correctly revealed or not. Moreover, we have recognized that there is a different intermetallic compound formation process between heating sources (electric heater and electric spark). We have studied in the previous research[27]. However, the reviewed manuscript lacked these explanatins as you pointed out, so we added the above explanatations in the revised manuscript line63.

Point 4:
Figure 10: SEM-BSE observation of reaction product can show other intermetallics in low amount.

Response 4:
Thank you for your comments. We can not identified other intermetallic compounds you pointed out.
To understand this opinion, we added following explanation in the revised manuscript line 226, what clear peaks of other intermetallic compounds have not been confirmed in the XRD analysis.

Reviewer 4 Report

The study reports an interesting investigation on determining the effect of particle size during aa self-propagating exothermic reaction. Few items are to be addressed before publication. The manuscript can be improved by considering the following: 

  • Please check the unit of particle size. Please replace ‘um’ with ‘µm’. Line 8
  • Please check the grammatical mistakes and sentence construction throughout the manuscript.
  • A detailed literature review is needed to bring out the need for the present work. Authors need to explain the limitations of current techniques in producing self-propagating exothermic reaction and the identified research gap to be addressed in this study.
  • A comprehensive review is needed to identify the gaps in the literature and novelty of this work. Available subject literature is to be explained in details by identifying the advantages, disadvantages and challenges with respect to industrial needs.
  • Authors need to clarify how many repeats were performed for each experimental conditions.
  • The scale bar within Fig. 3 (b) and (c) are not readable.
  • Authors are requested to report the temperature change in a tabulated format with standard deviation for each particle size in section 3.2.
  • In Figure 11, please use a different pattern of the legend to show the temperature variation. It’ll be easier to identify.
  • Please clarify the sentence starting in line 218: “As the exothermic ….”
  • Authors are requested to make comments on which particle size is more useful for industrial application and why is it so.
  • Please concise the conclusions point-wise identifying the key points.

Author Response

To Reviewer4
Thank you very much for your careful and fruitful reviewing. I deeply appreciate your suggestion and comments. We have revised the manuscript throughout. Please re-review the revised. Our replies are listed as follows:

Point 1:
Please check the unit of particle size. Please replace ‘um’ with ‘µm’. Line 8
Please check the grammatical mistakes and sentence construction throughout the manuscript.

Response 1:
Thank you for your pointed out.
We revised the correct units, ‘um’ to ‘µm’ in the revised manuscript.
I also asked a native speaker to check the English of the manuscript. The revised manuscript is attached. Please review it again.

Point 2:
A detailed literature review is needed to bring out the need for the present work. Authors need to explain the limitations of current techniques in producing self-propagating exothermic reaction and the identified research gap to be addressed in this study.
A comprehensive review is needed to identify the gaps in the literature and novelty of this work. Available subject literature is to be explained in details by identifying the advantages, disadvantages and challenges with respect to industrial needs.

Response 2:
As you pointed out, the goal of the experiments is not clear. We added the purpose and goal of the study and experiments in introduction section of the revised manuscript. Please see the attached.

Point 3:
Authors need to clarify how many repeats were performed for each experimental conditions.
The scale bar within Fig. 3 (b) and (c) are not readable.
Authors are requested to report the temperature change in a tabulated format with standard deviation for each particle size in section 3.2.

Response 3:
I apologize to reviewer what it was lack of explanations. I added error bars and related comments about variation of experimental data each necessary part of section.

Point 4:
In Figure 11, please use a different pattern of the legend to show the temperature variation. It’ll be easier to identify.

Response 4:
We are sorry. We modified marks in Fig. 11 to show the legend clearly.

Point 5:
Please clarify the sentence starting in line 218: “As the exothermic ….”
Authors are requested to make comments on which particle size is more useful for industrial application and why is it so.
Please concise the conclusions point-wise identifying the key points.

Response 5:
We revised the manuscript throughout. Particularly, in introduction section to be understood readers of the manuscript. These points you pointed out are expressed in the revised manuscript. Please see attached.

Round 2

Reviewer 1 Report

The revised manuscript showed significant improvement and thus it is recommended to be accepted in the present form.

Reviewer 2 Report

Accept in the present form